# Preoperative Evaluation and Surgical Simulation for Osteochondritis Dissecans of the Elbow Using Three-Dimensional MRI-CT Image Fusion Images

**DOI:** 10.3390/diagnostics11122337

**Published:** 2021-12-11

**Authors:** Sho Kohyama, Yasumasa Nishiura, Yuki Hara, Takeshi Ogawa, Akira Ikumi, Eriko Okano, Yasukazu Totoki, Yuichi Yoshii, Masashi Yamazaki

**Affiliations:** 1Department of Orthopaedic Surgery, Kikkoman General Hospital, Noda 278-0005, Japan; 2Tsuchiura Clinical Education and Training Center, Department of Orthopaedic Surgery, Tsukuba University Hospital, Tsuchiura 300-8585, Japan; ynishi@md.tsukuba.ac.jp; 3Department of Orthopaedic Surgery, Faculty of Medicine, University of Tsukuba, Tsukuba 305-8575, Japan; yukihara@md.tsukuba.ac.jp (Y.H.); eokano@tsukuba-seikei.jp (E.O.); yasutotoki@tsukuba-seikei.jp (Y.T.); masashiy@md.tsukuba.ac.jp (M.Y.); 4Mito Medical Center, Department of Orthopaedic Surgery, National Hospital Organization, Ibaraki 311-3193, Japan; ogawat@md.tsukuba.ac.jp; 5Mito Medical Education and Training Center, Department of Orthopaedic Surgery, Tsukuba University Hospital, Mito 310-0015, Japan; bravelupus193@gmail.com; 6Department of Orthopaedic Surgery, Tokyo Medical University Ibaraki Medical Center, Tokyo 300-0395, Japan; yyoshii@tokyo-med.ac.jp

**Keywords:** osteochondritis dissecans, elbow, magnetic resonance imaging, tomography, X-ray computed, imaging, three-dimensional, image interpretation, computer-assisted, simulation

## Abstract

We used our novel three-dimensional magnetic resonance imaging-computed tomography fusion images (3D MRI-CT fusion images; MCFIs) for detailed preoperative lesion evaluation and surgical simulation in osteochondritis dissecans (OCD) of the elbow. Herein, we introduce our procedure and report the findings of the assessment of its utility. We enrolled 16 men (mean age: 14.0 years) and performed preoperative MRI using 7 kg axial traction with a 3-Tesla imager and CT. Three-dimensional-MRI models of the humerus and articular cartilage and a 3D-CT model of the humerus were constructed. We created MCFIs using both models. We validated the findings obtained from the MCFIs and intraoperative findings using the following items: articular cartilage fissures and defects, articular surface deformities, vertical and horizontal lesion diameters, the International Cartilage Repair Society (ICRS) classification, and surgical procedures. The MCFIs accurately reproduced the lesions and correctly matched the ICRS classification in 93.5% of cases. Surgery was performed as simulated in all cases. Preoperatively measured lesion diameters exhibited no significant differences compared to the intraoperative measurements. MCFIs were useful in the evaluation of OCD lesions and detailed preoperative surgical simulation through accurate reproduction of 3D structural details of the lesions.

## 1. Introduction

Osteochondritis dissecans of the elbow (OCD) is a rare intra-articular osteochondral lesion that is associated with overhead throwing sports [1,2,3,4,5]. The subchondral bone and articular cartilage of the humeral capitellum are affected, and several possible causes, including repetitive microtrauma and genetic factors, are implicated [3,4,6,7]. Although the stability and size of the lesion are considered important factors impacting lesion severity [6,8,9,10], no single imaging modality can adequately predict lesion severity [11,12,13,14,15,16,17], thereby presenting a challenge to elbow surgeons. A recent study by Pu et al. concluded that a combination of radiography, computed tomography (CT), and magnetic resonance imaging (MRI) can most accurately determine OCD lesion stability by compensating for the respective flaws of the individual modalities [18]. However, at present, surgery is the only means of conclusively confirming the severity of OCD lesions using the classification proposed by the International Cartilage Repair Society (ICRS) [19,20,21,22,23].

Magnetic resonance arthrography (MRA) is another alternative modality for the evaluation of OCD lesions. As MRA involves the injection of contrast medium into the joint, the joint capsule distends, and visualization and differentiation of intra-articular structures can be enhanced [24,25]. Therefore, MRA could better depict articular cartilage. However, MRA is an invasive imaging modality and may cause pain, anxiety, and complications such as allergic reactions and infections [26,27]. As the majority of patients with OCD are children, it is important to minimize the use of invasive procedures. In addition, MRA alone cannot accurately evaluate the conditions of subchondral bone lesions such as sclerosis.

To address this difficulty, we developed and recently reported a method to create 3D MRI-CT fusion images (MCFIs) of the OCD lesions [28]. This computer-aided technique combines the advantages of CT and MRI and provides a minimally invasive, accurate preoperative evaluation of OCD lesions. In addition, detailed surgical simulation is possible, which could aid surgeons in intraoperative decision making.

There are various surgical options to effectively manage OCD lesions. In severe cases with unstable lesions, articular surface reconstruction must be considered. As the articular cartilage and subchondral lesion both require reconstruction, osteochondral autograft transplantation is a viable option. Generally, osteochondral autografts are harvested either from the knee using the osteochondral autograft transplantation system (OATS) or from the rib [1,6,11,29]. When the lesion is small but unstable, or the lesion is stable but resistant to conservative therapy, drilling of the lesion is an effective option. Drilling accelerates the union of the lesion and surrounding bone tissue by promoting bleeding from the bone marrow by puncturing the subchondral bone. If articular free bodies are present, their removal must be considered. Alternative surgical options include abrasion chondroplasty, microfracture, and in situ fixation of the lesion [1]. In this article, we introduce the use of MCFIs as a method of OCD lesion evaluation and surgical simulation for OCD lesions and report the findings of the assessment of the clinical applicability of the computer-aided technique. We aimed to assess the accuracy of MCFIs in evaluating OCD lesion severity and in facilitating surgical simulation.

## 2. Materials and Methods

### 2.1. Patient Selection

The institutional review board of the University of Tsukuba Hospital approved this study (Study Number: H29-58). Twenty-eight patients visited our facility and were diagnosed with OCD between July 2017 and March 2021. Among them, 16 patients whose lesions were evaluated using MCFIs were enrolled in this study. These patients subsequently underwent surgery. We obtained written informed consent from each patient. It was clearly stated that we would only use MCFIs for lesion evaluation and to decide the treatment strategy. All patients were boys (mean age: 14.0 ± 1.0 years, range: 12–16 years), and the right side was affected in fifteen patients, while the left side was affected in one patient. The average body weight of the patients was 56.9 (48.0–65.0) kg.

### 2.2. Obtaining the MR and CT Images

A 3-Tesla imager (MAGNETOM© Verio, Siemens, Munich, Germany) was used for MRI. We followed the procedures we published in a previous article on imaging sequence and settings, position of patients, and application of axial traction (7 kg) [28]. Axial traction widens the joint space and helps better visualize an outline of the articular cartilage of the humeral capitellum [28,30] (Figure 1).

We used a 320-row scanner (Aquilion ONE^TM^, Toshiba, Tokyo, Japan) for CT, and images were obtained with a 0.5 mm slice thickness. We did not apply axial traction during CT because CT data of the whole joint were not necessary for the procedure.

### 2.3. Creation of 3D Models

Using the obtained data, we created 3D MRI models of the humerus and articular cartilage and a 3D CT model of the humerus. The Materialise Mimics Innovation Suite version 20 (Materialise©, Leuven, Belgium) was used for this procedure. We referred to the MR signal intensity of the articular cartilage and humerus in each case for creation of 3D MRI models. As the participants in this study were skeletally immature, the growth cartilage was present in some cases. First, we set a threshold for the MR signal intensity for each target tissue. According to the set threshold, we selected the pixels that corresponded to the target tissue. The threshold was roughly set automatically and adjusted manually while simultaneously referring to the monitor to ensure correct selection of the target tissue. This procedure is called segmentation, which is crucial for creating better images. While segmenting the articular cartilage, we defined the articular cartilage fissures (ACFs) as the low-intensity lines within the articular cartilage, which penetrate or are perpendicular to the articular surface [12,15,31,32,33]. Articular surface deformity (ASD) was defined as irregularities in the outline of the articular cartilage [12,15] (Figure 2). The segmented structures, as well as the ACFs and ASDs were reconstructed and reproduced into 3D models. Similarly, we created 3D CT models of the humerus. During the procedure, the subchondral bone was considered segmented when the discontinuity of the subchondral bone to the floor was observed in all three planes: axial, sagittal, and coronal. We manually created a separate 3D model of segmented subchondral bone (SSB) and displayed it in red color for better visualization (Figure 3).

### 2.4. Fusion of Created 3D Models

We used the 3-matic software version 12 (Materialise©, Belgium) to fuse the 3D models. First, we exported the created 3D models from the Materialise Mimics to the 3-matic. We then roughly fused both the 3D MRI and 3D CT models of the humerus using a function called N-point registration. This function enables two separate 3D models to be superimposed using an arbitrary number of corresponding points. In our procedure, we registered four corresponding points that are easy to recognize and belong to different planes, as reported in a previous study [28] (Figure 4a). Second, we used a function called global registration to fine-tune the position of the superimposed 3D models (Figure 4b). Using this function, the positions of the aligned 3D models can be automatically corrected depending on their shapes. Throughout the procedures, the 3D MRI model of the articular cartilage was set to move together with the 3D MRI model of the humerus in order to maintain the positional relationship between the structures. Finally, we completed the fusion of the 3D CT model of the humerus and the 3D MRI model of the articular cartilage by hiding the 3D MRI model of the humerus (Figure 4c). We termed this fusion model MCFI. The first author, an elbow surgeon with 14 years of clinical experience, created all MCFIs. The average interval between the MCFI creation and surgery was 26 (5–70) days.

### 2.5. Lesion Evaluation Using MCFIs

The first author evaluated the OCD lesion immediately after image creation based on the MCFI. The evaluation was particularly focused on the findings of the articular cartilage and subchondral bone, such as the presence of ACFs, articular cartilage defect (ACD), or ASD (Figure 5a), and whether the subchondral bone was segmented or not. We recorded the findings of each case. In order to differentiate ASDs from ACDs, deformities were recorded as either protrusions or flattening of the articular surface. The positional relationship between the ACF and the SSB was evaluated by adjusting the transparency of the articular cartilage (Figure 5b). The articular surface of the humeral capitellum was considered elliptical in the anteroposterior view of the MCFI and divided into four areas, areas 1 to 4, clockwise from the anteromedial area (Figure 5c); we thereby recorded the location of each finding. Based on the findings reproduced in the MCFI, the vertical and horizontal diameters of the lesion were also measured (Figure 5d).

OCD lesions are defined as unstable when the lesion is displaced or dislocated, and thereby the articular surface is deformed [12,14,15,16,32,33,34]. We defined unstable OCD lesions in MCFIs as those with ACFs and/or ASDs, along with the SSB underneath.

Based on the created MCFI, the ICRS classification [20] was predicted independently and comprehensively by two elbow surgeons (assessors 2 and 3 who had 13 and 12 years of clinical experience, respectively). The surgeons were requested to predict the ICRS classification at least 3 days preceding the operation. The surgeons were cognizant of the ICRS classification and were blinded to the patients’ medical records. The two assessors participated as assistant surgeons in several surgeries performed in this study.

### 2.6. Surgical Simulation

We used the 3-matic software version 12 (Materialise©, Leuven, Belgium) for the surgical simulation. The first author performed all surgical simulations in this study.

Articular surface reconstruction by costal osteochondral autograft transplantation [19] is our first choice when the OCD lesion is unstable and its maximum diameter exceeds 10 mm; thus, we simulated the surgical procedure. First, we decided on the resection area according to the reproduced ACFs and ASDs (Figure 5d). After selecting the resection area, we separated the area from the original 3D model of the articular cartilage and created an independent 3D model. This enabled us to freely hide the resection area, simulating the lesion resection (Figure 5e). Second, we simulated autograft transplantation. A 3D model of the average-sized costal osteochondral autograft was created in advance (Figure 6). We created this model based on the size of the actual autograft harvested from previously operated cases. This autograft model can be freely placed within the MCFI, and we simulated the position, direction, and depth of the transplantation (Figure 7).

We simulated drilling when the OCD lesion was ICRS class IV and small, or when the lesion was stable, articular surface reconstruction was not necessary, and the lesion did not respond to conservative therapy. Generally, subchondral OCD lesions with a chronic history have a sclerotic component [1]. It is essential to penetrate the subchondral sclerosis to promote bone healing. Therefore, as the first step of the simulation of drilling, we manually selected the sclerotic region and highlighted it with a different color for better visualization. Second, we simulated the entry point, direction, and depth of drilling. When the lesion is stable and the articular surface is intact, it is important to avoid iatrogenic articular cartilage damage. Therefore, we simulated posteroanterior drilling for those cases (Figure 8).

### 2.7. Intraoperative Evaluation of ICRS Classification

We evaluated all OCD lesions intraoperatively either under direct observation or using arthroscopic examination to determine the ICRS classification [20]. According to the definition, class I and II lesions are stable, while class III and IV lesions are unstable.

### 2.8. Evaluation of MCFIs

We compared the predicted values with the actual intraoperative findings for the ACFs, ACDs, ASDs, vertical and horizontal lesion diameters, and ICRS classification. The corresponding rate between each finding determined from the MCFI and those determined intraoperatively were evaluated. We calculated the corresponding rate as follows: (number of cases in which the findings determined by the MCFI corresponded to the actual intraoperative findings)/(total number of cases) [17].

Each finding and the corresponding ICRS classification were intraoperatively determined by both visual inspection and palpation. In class III lesions that underwent articular reconstruction, the diameter of the resected lesion was recorded. In class IV lesions, the diameters of the articular cartilage defect were recorded. In 9 of 16 cases, where the first author conducted the surgery, another elbow surgeon with 23 years of clinical experience evaluated the intraoperative findings. This surgeon did not evaluate the lesion using the MCFI. The primary surgeon assessed the intraoperative findings in the remaining seven cases. Among the seven cases, six cases were assessed by an elbow surgeon with 23 years of clinical experience and one case by an elbow surgeon with 34 years of clinical experience. We compared the predicted and intraoperatively measured vertical and horizontal lesion diameters using the Mann–Whitney U test, which was selected because of the small sample size. Statistical significance was set at *p* < 0.05.

## 3. Results

### 3.1. Lesion Evaluation Using MCFIs

Table 1, Table 2 and Table 3 show the findings predicted by the MCFIs and intraoperatively determined findings for the ACFs, ACDs, and ASDs, respectively. The reproducibility of each finding was accurate in the MCFI, except for one case, in which the lesion was detached at the time of surgery, resulting in a corresponding rate of 93.8%. Table 4 depicts the measurements of the diameters of the lesions. The median values of the vertical diameter measured preoperatively and intraoperatively were 14.8 and 14.0 mm, respectively; there was no significant difference (*p* = 0.78). The median values of the horizontal diameter measured preoperatively and intraoperatively were 13.1 and 12.0 mm, respectively; there was no significant difference (*p* = 0.14). Figure 9 shows the intraoperative findings and corresponding MCFIs for representative cases.

### 3.2. ICRS Classifications

Table 5 presents the predicted and intraoperative ICRS classifications. The MCFI resulted in a corresponding rate of 93.8% for both examiners. This is because there was one case in which the lesion was detached at the time of surgery (case 15). We predicted this case as ICRS class III; however, the case was confirmed as class IV intraoperatively.

### 3.3. Surgical Simulation

We simulated drilling in four cases and osteochondral autograft transplantation in eleven cases. One case was indicated to perform free-body removal alone. All surgeries were conducted based on the simulations. We performed drilling for cases 1, 3, 11, and 15, free-body removal for cases 6 and 15, and costal osteochondral autograft transplantation for the remaining cases. Based on the MCFI, case 15 was indicated for articular surface reconstruction, but the patient did not consent to the surgical procedure and opted for the drilling procedure. We conducted all surgeries as simulated in all cases. Figure 10 shows the intraoperative findings and corresponding surgical simulation for representative cases.

## 4. Discussion

OCD lesion evaluation using MCFIs is a minimally invasive technique that enables prediction of lesion severity with high accuracy [28]. In developing this technique, we intended to maximize the advantages and compensate for the shortcomings of MRI and CT. The application of axial traction widens the joint space of the radio-capitellar joint and improves visualization of the articular cartilage outline of the humeral capitellum with minimum pain and discomfort [30,31], making the creation of accurate 3D models of the articular cartilage possible, and enabling images with a slice thickness of 0.4 mm to be obtained using a 3D sequence. We used a 7 kg traction weight according to previous studies [30,31]. There are no studies clarifying the ideal traction weight for elbow MRI in a skeletally immature population. Ideally, the traction weight must be decided on the basis of the body weight, size, and muscle development of the patient. Therefore, in the future, we will attempt to determine the ideal traction weight to lower discomfort during application of traction during MRI as much as possible.

The assessor can evaluate the 3D structure of the lesion from arbitrary angles using the MCFI. Additionally, it is possible to precisely obtain a positional relationship between the articular cartilage and the subchondral bone by adjusting the transparency of the articular cartilage, which is not visible to surgeons even during surgery. We accurately predicted the lesion severity in 15 out of 16 cases (93.8% accuracy). Although we predicted the case 15 lesion as ICRS class III, the lesion was diagnosed as class IV intraoperatively. This may be due to the high instability of the lesion, which could have caused the lesion to be displaced from the floor sometime between image acquisition and surgery. However, regarding the detection of unstable lesions, our technique achieved 100% accuracy. Diagnosing the stability of the lesion is extremely important in determining treatment strategies for patients with OCD [1,6,20]. Therefore, we believe that our technique is highly effective.

We simulated the surgical procedure in all cases and conducted surgery per the simulations in each case, which is another advantage of the MCFI. Among the surgical procedures we performed, articular surface reconstruction using a costal osteochondral autograft is the most complex surgery. Poor reconstruction of the articular surface leads to osteoarthritis in the future. However, owing to the anatomical feature of the elbow joint [35], articular surface reconstruction referring only to intraoperative findings can be challenging in some cases. Detailed surgical planning using MCFIs has the potential to minimize the process of intraoperative decision making by surgeons.

We selected costal osteochondral autograft transplantation for articular surface reconstruction because we are accustomed to the procedure; however, some surgeons prefer to harvest the osteochondral autograft from the knee using the OATS technique. Our surgical simulation technique is also applicable to surgeries that incorporate the OATS technique. When simulating the procedure, it is necessary to prepare a 3D model of a cylindrical autograft resembling the autograft harvested using the OATS technique instead of the 3D model of the costal osteochondral autograft. By preparing cylindrical autografts of various diameters, surgeons can precisely simulate the location and direction of the transplant, as well as the number of autografts necessary for the procedure (Figure 11).

A limitation of this technique is that the segmentation of the articular cartilage is time-consuming and technically demanding. Precise segmentation of the articular cartilage is the most crucial part of the procedure, but it must be partly executed manually. No available software can automatically and completely segment the articular cartilage, and it takes approximately 2 h per case to complete the procedure. Because we rely on the created MCFIs to determine the treatment strategies, it takes several weeks from the image acquisition to the surgery. This delay may lead to the deterioration of the OCD lesion in some cases, such as case 15, in which we misdiagnosed the ICRS classification. Therefore, we need to simplify the procedure and minimize the waiting period to avoid such consequences. In this study, evaluation of intraoperative findings was subjectively performed by each surgeon. We hope to achieve objective superposition of the MCFI and the surgical field in the future. Another limitation is that the number of cases was limited. The accumulation of cases takes time as OCD is a rare condition. We will continue to work on the current study to further evaluate the consistency between intraoperative findings and MCFIs. Increasing the number of cases could lead to the development of software that can automatically create the MCFI for OCD patients. We also aim to conduct more quantitative evaluations of OCD lesions, including assessment of mild cases to predict the progression of lesion severity, in the future. In addition, we hope to apply this technique to other joints, with the aim of assisting surgeons in various fields treating intra-articular osteochondral lesions.

## 5. Conclusions

Herein, we introduced OCD lesion evaluation and surgical simulation using MCFIs of OCD, a novel tool developed by us. This computer-aided technique made the accurate evaluation of 3D structural details of the articular cartilage and subchondral bone possible, as well as lesion stability, without surgical intervention. A detailed surgical simulation is potentially useful for minimizing intraoperative decision making.

## 6. Patents

A patent for the 3D MRI-CT fusion image construction technique described in this article was applied for in Japan; 3D model generation method, 3D model generation device, and 3D model-generation program. Application No. 2018-066054, pending approval.

## Figures and Tables

**Figure 1 diagnostics-11-02337-f001:**
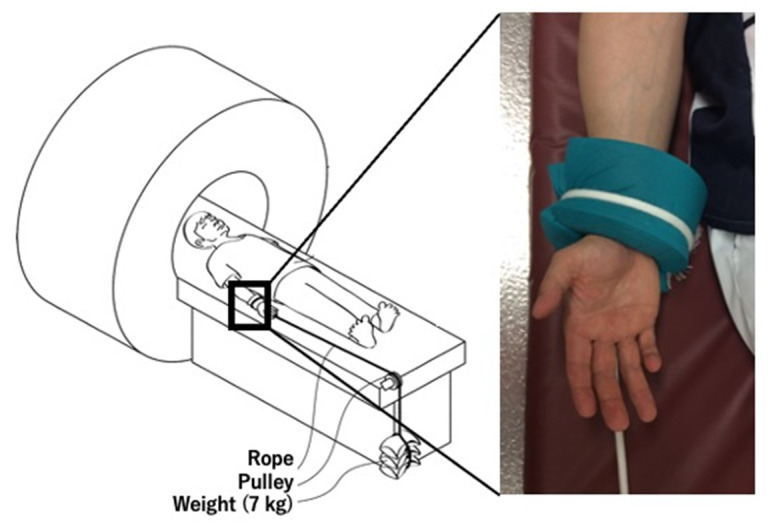
Acquisition of magnetic resonance images with axial traction. Axial traction (7 kg) was applied to the elbow to better visualize the articular cartilage of the humeral capitellum.

**Figure 2 diagnostics-11-02337-f002:**
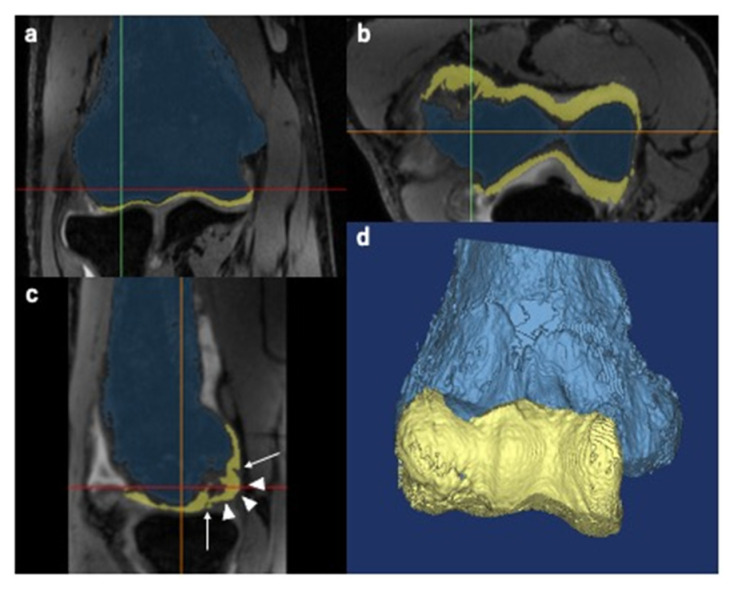
Creation of three-dimensional (3D) magnetic resonance imaging (MRI) models. The pixels correspond to the humerus and the articular cartilage which were independently selected based on the set threshold. The blue color represents the humerus, and the yellow color represents the articular cartilage. Green, red and orange lines are the reference lines correspond to sagittal, axial and coronal, respectively. (**a**) Coronal view. (**b**) Axial view. (**c**) Sagittal view. Arrow: articular cartilage fissures. Arrowheads: articular surface deformities. (**d**) Reconstructed 3D image of the humerus and articular cartilage.

**Figure 3 diagnostics-11-02337-f003:**
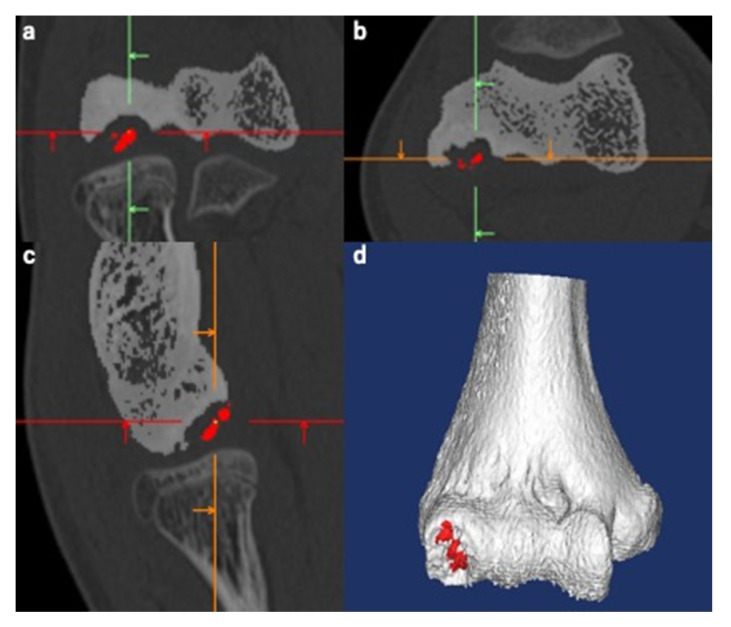
Creation of a three-dimensional (3D) computed tomography (CT) model of the humerus. The same procedure as used with magnetic resonance imaging was used. The segmented subchondral bone (SSB) was defined as the lesion whose continuity to the floor was lost in all three planes. The separate SSB model was created manually and is displayed in red for better visualization. Green, red and orange lines are the reference lines correspond to sagittal, axial and coronal, respectively. (**a**) Coronal view. (**b**) Axial view. (**c**) Sagittal view. (**d**) Reconstructed 3D image of the humerus.

**Figure 4 diagnostics-11-02337-f004:**
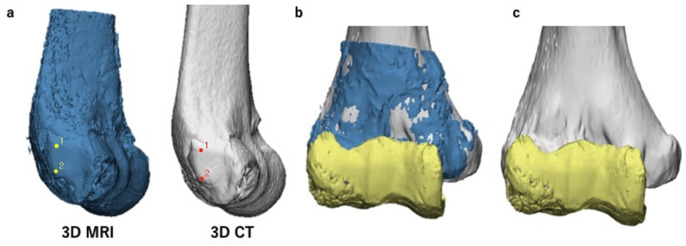
Procedures to create magnetic resonance imaging-computed tomography fusion images (MCFIs). The three-dimensional (3D) magnetic resonance imaging (MRI) model of the articular cartilage and 3D computed tomography (CT) model of the humerus were fused. (**a**) N-point registration of the 3D MRI and 3D CT models of the humerus for rough superimposition. The marked points in this figure are the two protuberances of the lateral epicondyle. (**b**) A superimposed 3D model after global registration (anteroposterior view). Here, the 3D CT and 3D MRI models of the humerus and the 3D MRI model of the articular cartilage are shown in gray, blue, and yellow, respectively. (**c**) A complete MCFI (anteroposterior view). The 3D MRI model of the humerus was hidden. This completed the creation of MCFIs.

**Figure 5 diagnostics-11-02337-f005:**
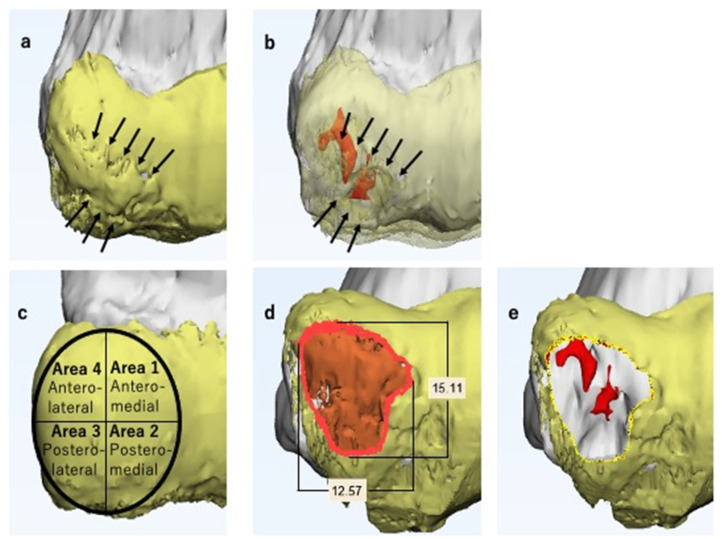
Magnetic resonance imaging-computed tomography fusion images (MCFIs). Here, the 3D CT model of the humerus and the 3D MRI model of the articular cartilage are shown in gray and yellow, respectively. (**a**) Anteroposterior view of the lesion (enlarged). The articular cartilage fissures (ACFs) can be observed (arrows). The articular surface is protruded and not smooth, particularly in the area surrounded by the fissure. (**b**) Anteroposterior view of the lesion with a transparent articular cartilage (enlarged). The segmented subchondral bone is shown in red and is located beneath the fissure (arrows). Considering all the findings, we predicted the lesion as unstable. (**c**) On the anteroposterior view of the MCFI, we divided the surface into four areas and recorded the location of each finding. (**d**) The lesion is marked in orange. The vertical and horizontal diameters of the lesion were measured. We determined the size of the lesion comprehensively based on the findings of the articular cartilage and subchondral lesion. (**e**) Simulated image of the lesion resection. We separated the area selected in Figure 5d from the original three-dimensional magnetic resonance imaging articular cartilage model and hid it to simulate lesion selection.

**Figure 6 diagnostics-11-02337-f006:**
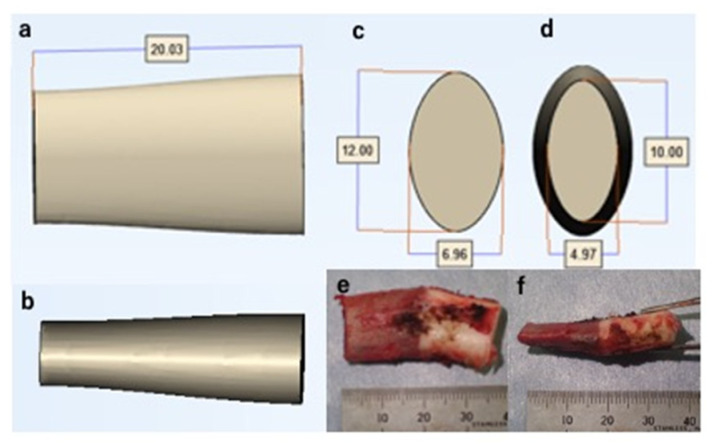
A three-dimensional model of an average-sized costal osteochondral autograft. (**a**) Anteroposterior view. (**b**) Lateral view. (**c**) Distal view. (**d**) Proximal view. (**e**) Anteroposterior view of a harvested costal osteochondral autograft. (**f**) Lateral view of a harvested costal osteochondral autograft.

**Figure 7 diagnostics-11-02337-f007:**
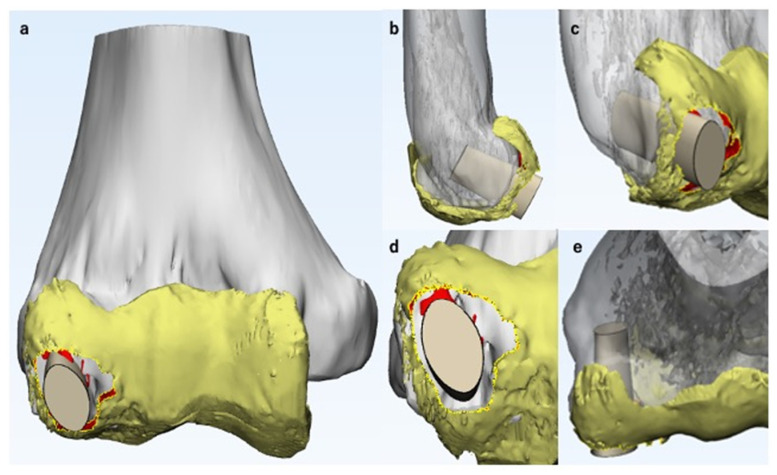
Simulation of costal osteochondral autograft transplantation. Here, the 3D CT model of the humerus and the 3D MRI model of the articular cartilage are shown in gray and yellow, respectively. The three-dimensional model of the costal osteochondral autograft was placed on the magnetic resonance imaging-computed tomography fusion image to simulate the position, direction, and depth of the transplant. (**a**) Anteroposterior view. (**b**) Lateral view. (**c**) Oblique view. (**d**) Distoproximal view. (**e**) Proximodistal view.

**Figure 8 diagnostics-11-02337-f008:**
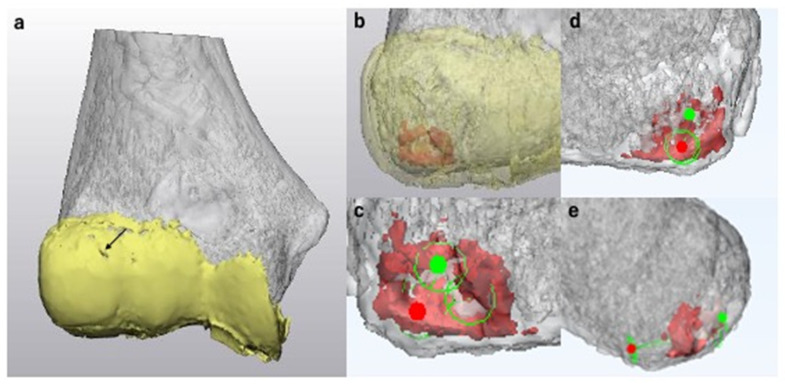
Surgical simulation of drilling. Here, the 3D CT model of the humerus and the 3D MRI model of the articular cartilage are shown in gray and yellow, respectively. (**a**) An anteroposterior view of the magnetic resonance imaging-computed tomography fusion image (MCFI) of case 1. The articular surface was smooth with only a small fissure in area 1 (arrow). (**b**) MCFI with a transparent articular cartilage. The subchondral lesion with sclerosis is shown in red. (**c**) Simulation of drilling (enlarged anteroposterior view without the articular cartilage). We simulated to penetrate the spots with sclerosis and cyst formation. The entry point is shown as a red spot, and the target point is shown in green. (**d**) Posteroanterior view. To avoid iatrogenic articular cartilage damage, we simulated posteroanterior drilling in these cases. The green circle with a red spot represents the entry point for drilling. (**e**) Lateral view. We simulated the direction and depth of drilling.

**Figure 9 diagnostics-11-02337-f009:**
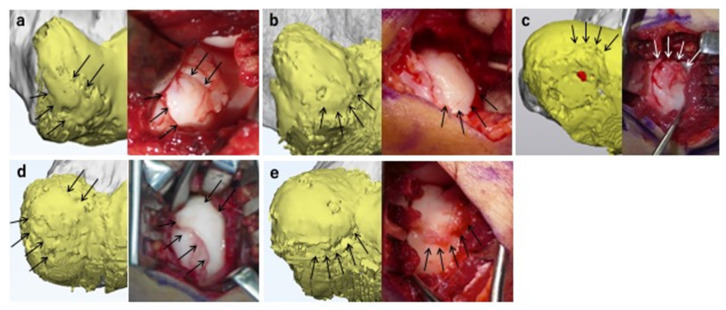
Magnetic resonance imaging-computed tomography fusion images (MCFIs) and corresponding intraoperative findings. The MCFIs are shown in the left panels, and the corresponding intraoperative findings are shown in the right panels. Here, the 3D CT model of the humerus and the 3D MRI model of the articular cartilage are shown in gray and yellow, respectively. (**a**) Case 12. The MCFI accurately reproduced the articular cartilage fissures (ACFs) in areas 1, 2, 3, and 4 (arrows). The lesion was predicted as unstable because the segmented subchondral bone (SSB) was present underneath the ACFs, and the articular surface was protruded. As predicted, the lesion was classified as unstable intraoperatively. (**b**) Case 8. The MCFI correctly reproduced the protrusion of the articular surface in areas 2 and 3 (arrows). The lesion was predicted as unstable owing to the presence of the articular surface deformity (ASD) and SSB underneath the ACF. The lesion was unstable on palpation, as predicted. (**c**) Case 16. The MCFI correctly reproduced the ACF and ASD in areas 2 and 3 (arrows). The lesion was predicted as unstable based on these findings and intraoperatively classified that the lesion was unstable. (**d**) Case 9. The MCFI correctly reproduced the ACFs in areas 1, 3, and 4 (arrows). The lesion was predicted as unstable based on the presence of the SSB underneath the ACF, as well as ASD. The lesion was unstable intraoperatively on palpation, as predicted. (**e**) Case 5. The MCFI correctly reproduced the ACF in areas 2 and 3 (arrows). The lesion was predicted as unstable because of the presence of ACF and the SSB underneath and ASD. The lesion was classified as unstable on palpation.

**Figure 10 diagnostics-11-02337-f010:**
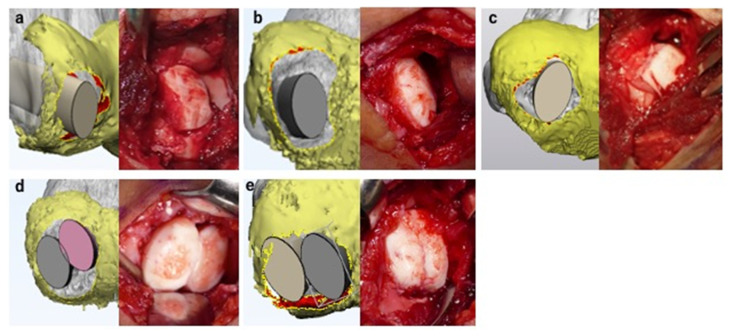
Surgical simulation using the magnetic resonance imaging-computed tomography fusion images and corresponding intraoperative findings. The surgical simulations are shown in the left panel, and corresponding intraoperative findings are shown in the right panel. Costal osteochondral autograft transplantation was simulated in all presented cases. All surgeries were conducted as simulated. Here, the 3D CT model of the humerus and the 3D MRI model of the articular cartilage are shown in gray and yellow, respectively. (**a**) Case 12. Reconstruction of the articular surface and lateral wall of the capitellum was necessary for this case. (**b**) Case 8. Reconstruction of the articular surface and lateral wall of the capitellum was necessary for this case. (**c**) Case 16. One costal osteochondral autograft transplantation procedure was sufficient to cover the articular surface defect. (**d**) Case 9. The predicted lesion was large; therefore, we simulated reconstruction of the articular surface by two costal osteochondral autografts to cover as much surface as possible. (**e**) Case 5. Two costal osteochondral autografts were necessary to reconstruct the large, predicted lesion.

**Figure 11 diagnostics-11-02337-f011:**
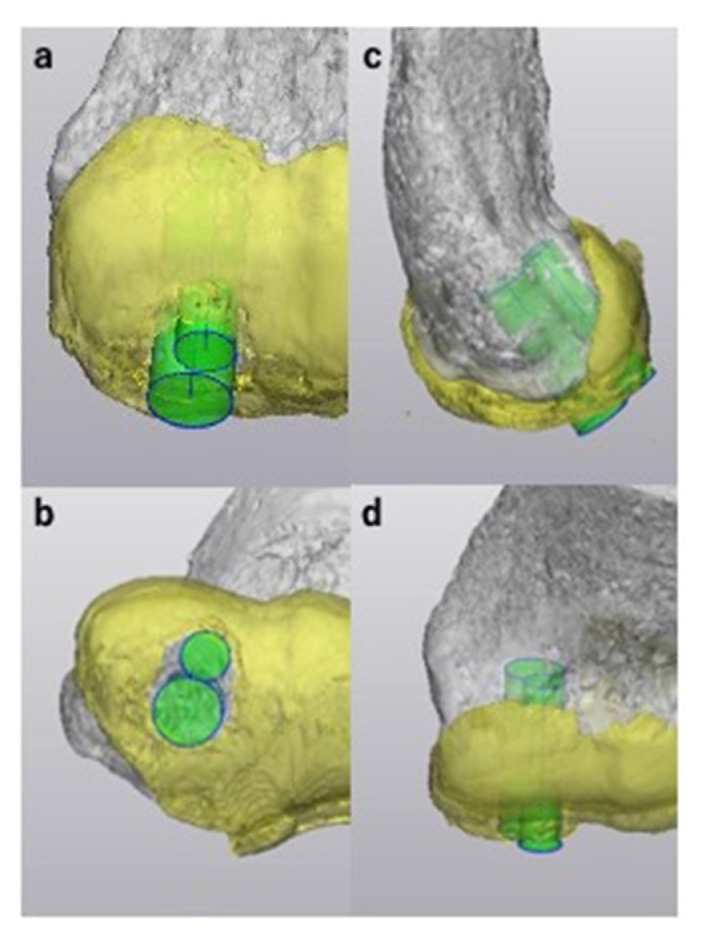
Sample simulation of articular surface reconstruction using the osteochondral autograft transplantation system technique with three-dimensional (3D) models of cylindrical autografts for case 16. We created two 3D models of cylindrical autografts with diameters of 5 mm and 7 mm. The 3D models were positioned to cover the lesion to simulate the location, direction, and depth of the transplant. Here, the 3D CT model of the humerus, the 3D MRI model of the articular cartilage and 3D models of cylindrical autografts are shown in gray, yellow and green, respectively. (**a**) Anteroposterior view. (**b**) Distoproximal view. (**c**) Lateral view. (**d**) Proximodistal view.

**Table 1 diagnostics-11-02337-t001:** Validation of the three-dimensional magnetic resonance imaging-computed tomography fusion images (3D MRI-CT fusion images; MCFIs) against intraoperative findings: articular cartilage fissures (ACFs). The reproducibility of the ACFs by the MCFIs was accurate in all cases.

Cases	Localization of Articular Cartilage Fissures (If Present)	
	Findings from 3D MRI-CT Fusion Images	Intraoperative Findings
1	Area 1	Area 1
2	Areas 1, 2, 3, and 4	Areas 1, 2, 3, and 4
3	None	None
4	Area 2	Area 2
5	Areas 1, 2, and 3	Areas 1, 2, and 3
6	None	None
7	Area 2	Area 2
8	Areas 2 and 3	Areas 2 and 3
9	Areas 1, 3, and 4	Areas 1, 3, and 4
10	Areas 1 and 4	Areas 1 and 4
11	Area 1	Area 1
12	Areas 1, 2, 3, and 4	Areas 1, 2, 3, and 4
13	Areas 2 and 3	Areas 2 and 3
14	Areas 2 and 3	Areas 2 and 3
15	Areas 1, 2, 3, and 4	Areas 1, 2, 3, and 4
16	Areas 2 and 3	Areas 2 and 3

**Table 2 diagnostics-11-02337-t002:** Validation of the magnetic resonance imaging-computed tomography fusion image (MCFI) against intraoperative findings: articular cartilage defects (ACDs). The reproducibility of the ACDs by MCFIs was accurate except for case 15, whose capitellar osteochondritis lesion was detached from the floor at the time of surgery.

Cases	Localization of Articular Cartilage Defects (If Present)	
	Findings from 3D MRI-CT Fusion Images	Intraoperative Findings
1	None	None
2	None	None
3	Areas 2 and 3	Areas 2 and 3
4	Areas 2 and 3	Areas 2 and 3
5	None	None
6	Area 2	Area 2
7	Areas 2 and 3	Areas 2 and 3
8	None	None
9	None	None
10	Area 1	Area 1
11	Area 1	Area 1
12	None	None
13	None	None
14	None	None
15	None	Areas 1, 2, 3, and 4 Lesion detached
16	None	None

**Table 3 diagnostics-11-02337-t003:** Validation of the magnetic resonance imaging-computed tomography fusion images (MCFIs) against intraoperative findings: articular surface deformities (ASDs). The MCFIs precisely reproduced the ASDs in all cases except for case 15. The lesion appeared to be protruded on the MCFI but was detached from the floor at the time of surgery.

Cases	Localization of Articular Surface Deformities (If Present)	
	Findings from 3D MRI-CT Fusion Images	Intraoperative Findings
1	None	None
2	Areas 2 and 3 (protrusion)	Areas 2 and 3 (protrusion)
3	None	None
4	None	None
5	Areas 2 and 3 (protrusion)	Areas 2 and 3 (protrusion)
6	None	None
7	None	None
8	Areas 2 and 3 (protrusion)	Areas 2 and 3 (protrusion)
9	Areas 2 and 3 (protrusion)	Areas 2 and 3 (protrusion)
10	Areas 1, 2, 3, and 4 (flattening)	Areas 1, 2, 3, and 4 (flattening)
11	Areas 1, 2, 3, and 4 (flattening)	Areas 1, 2, 3, and 4 (flattening)
12	Areas 2 and 3 (protrusion)	Areas 2 and 3 (protrusion)
13	Areas 2 and 3 (protrusion)	Areas 2 and 3 (protrusion)
14	Areas 2 and 3 (protrusion)	Areas 2 and 3 (protrusion)
15	Areas 1, 2, 3, and 4 (protrusion)	Areas 1, 2, 3, and 4 (lesion detached)
16	Areas 2 and 3 (protrusion)	Areas 2 and 3 (protrusion)

**Table 4 diagnostics-11-02337-t004:** Validation of the magnetic resonance imaging-computed tomography fusion images (MCFIs) against intraoperative findings: vertical and horizontal lesion diameters. The reproducibility of the vertical and horizontal diameters of the lesions was accurate in the MCFI. In case 1, there were no articular cartilage defects or articular surface deformity. Therefore, it was impossible to measure the lesion size.

Cases	Vertical Diameters (mm)		Horizontal Diameters (mm)	
	MCFI	Intraoperative Findings	MDFIs	Intraoperative Findings
1				
2	19.7	18	12.3	12
3	7.9	8	11.5	11
4	13.6	12	14.5	14
5	16.1	14	15.6	15
6	11.6	10	8.8	8
7	12.1	15	14.3	13
8	18.7	17	12.5	12
9	19.4	19	14.1	14
10	16.9	16	14.7	12
11	19.5	20	16	15
12	16	18	12	12
13	11.6	11	7.8	8
14	14.8	14	11.8	10
15	13.5	13	14.2	11
16	12.5	12	13.1	10
Median value (interquartile range)	14.8 (12.1–16.9)	14.0 (11.5–17.5)	13.1 (11.8–14.3)	12.0 (10–14)
*p*-value	0.78		0.14	

**Table 5 diagnostics-11-02337-t005:** Comparison of the International Cartilage Repair Society (ICRS) classification predicted by examiners and the actual intraoperative findings. The intraoperative ICRS classification accurately corresponded to the magnetic resonance imaging-computed tomography fusion image-based predictions in all cases except for case 15. The match rate was 93.8%.

Cases	MCFI		Intraoperative Findings
	Assessor 2	Assessor 3	
1	II	II	II
2	III	III	III
3	IV	IV	IV
4	IV	IV	IV
5	III	III	III
6	IV	IV	IV
7	IV	IV	IV
8	III	III	III
9	III	III	III
10	IV	IV	IV
11	IV	IV	IV
12	III	III	III
13	III	III	III
14	III	III	III
15	III	III	IV
16	III	III	III

## Data Availability

Not applicable.

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
