# Peer review of "Preoperative Evaluation and Surgical Simulation for Osteochondritis Dissecans of the Elbow Using Three-Dimensional MRI-CT Image Fusion Images"

_diagnostics, 2021, doi:10.3390/diagnostics11122337_

Round 1

Reviewer 1 Report

The enhanced view of the osteochondritis dissecans can be obtained using 
MR contrast arthrography (MRA) imaging shows the OCD lesion dissection and/or dissected fragment separation much better. 
The authors should explain why they omitted the MRA imaging by pointing to two imaging modalities instead of a single MRA. 
Authors present data from the young population of patients the average age is 14 years, so children and adolescents were in the study group author should mention that using term "boys" instead of "men."
Authors should clearly describe the side of the lesions instead of writing that one patient had affected the left side.
Applying traction and its weight during the exam should instead address the body weight. First of all, it is well accepted in the children population. The traction weight may be different for patients with different body weights, sizes and muscles development. 
The exact weight of traction may produce joint distention in some cases and may lead to strain in some weaker cases. Authors use the term articular cartilage fissures (ACF) that may be confusing the fissure may have the meaning of the damage deriving. Probably the joint or the articular space should fit better to the text. 
One should point out the skeletally immature participants, and the growth cartilage is present at that age.

Author Response

Authors’ Reply to the Review Report (Reviewer 1)

Reviewer 1

We would like to thank the reviewer for the critique of our manuscript and insightful and constructive comments, which have helped improve the manuscript. We have made every effort to address the issues raised by the reviewer. The revisions are highlighted in the revised manuscript. Please find the detailed responses to the reviewer's comments in a point-by-point fashion below. We hope that the revisions made by us address the concerns raised by you adequately.

The enhanced view of the osteochondritis dissecans can be obtained using MR contrast arthrography (MRA) imaging shows the OCD lesion dissection and/or dissected fragment separation much better. 
The authors should explain why they omitted the MRA imaging by pointing to two imaging modalities instead of a single MRA. 

⇨ Thank you for your constructive comment. As pointed out by you, MRA, which involves injection of contrast medium into the joint, can depict articular cartilage better than does conventional MRI. However, as the majority of patients with OCD are children, it is important to minimize the use of invasive procedures. In addition, MRA alone cannot accurately evaluate subchondral bone OCD lesions. We have added the following paragraph in the Introduction section (lines 52-60):

“Magnetic resonance arthrography (MRA) is another alternative modality for the evaluation of OCD lesions. As MRA involves injection of contrast medium into the joint, the joint capsule distends and visualization and differentiation of intra-articular structures can be enhanced [24,25]. Therefore, MRA could better depict articular cartilage. However, MRA is an invasive imaging modality and may cause pain, anxiety, and complications such as allergic reactions and infections [26,27]. As the majority of patients with OCD are children, it is important to minimize the use of invasive procedures. In addition, MRA alone cannot accurately evaluate conditions related to subchondral bone lesions such as sclerosis.”

Authors present data from the young population of patients the average age is 14 years, so children and adolescents were in the study group author should mention that using term "boys" instead of "men."

Authors should clearly describe the side of the lesions instead of writing that one patient had affected the left side.

⇨ Thank you for your suggestion. We have revised the text in the Patient Selection subsection as follows (lines 89-91):

“All patients were boys (mean age: 14.0 ± 1.0 years, range: 12–16 years), and the right side was affected in fifteen patients and the left side was affected in one patient.”

Applying traction and its weight during the exam should instead address the body weight.

⇨ Thank you for your suggestion. We have added the following sentence in the Patient Selection subsection (line 91): “The average body weight of the patients was 56.9 (48.0–65.0) kg.”

First of all, it is well accepted in the children population. The traction weight may be different for patients with different body weights, sizes and muscles development. 
The exact weight of traction may produce joint distention in some cases and may lead to strain in some weaker cases.

⇨ Thank you for pointing this out. As pointed out by you, the ideal traction weight may be different for different patients and may depend on the patient’s body size. However, there are no previous studies investigating the ideal traction weight for elbow MRI in a skeletally immature population. This is an important limitation of this study; therefore, we have added the following text in the Discussion section (lines 353-358):

“We used a 7-kg traction weight according to previous studies [30, 31]. There are no studies clarifying the ideal traction weight for elbow MRI in a skeletally immature population. Ideally, the traction weight must be decided on the basis of the body weight, size and muscle development of the patient. Therefore, in the future, we will attempt to determine the ideal traction weight to lower discomfort during application of traction during MRI as much as possible.”

Authors use the term articular cartilage fissures (ACF) that may be confusing the fissure may have the meaning of the damage deriving. Probably the joint or the articular space should fit better to the text. 

⇨ Thank you for your comment. Here, ACF is not either the joint space or the articular space. As defined in lines 115-117, the low-intensity lines within the articular cartilage, which penetrate or are perpendicular to the articular surface are defined as ACF. ACF in the MCFI is composed of several MRI slices with these low-intensity lines within the articular cartilage. Therefore, we believe the definition of ACF is correct as is. To clarify this, we have added a sentence in lines 118-120. Thank you.

“The segmented structures, as well as ACF and ASD were reconstructed and reproduced into 3D models.”

One should point out the skeletally immature participants, and the growth cartilage is present at that age.

⇨ Thank you for your comment. Per your comment, we have added the following sentence in lines 109-110:

“As the participants in this study were skeletally immature, the growth cartilage was present in some cases.”

Reviewer 2 Report

Osteochondritis dissecans (OCD) is a specific kind of pathology and no single imaging modality can adequately predict lesion severity thereby presenting a challenge to elbow surgeons. Though the authors address a rare pathology, the overall results could be transferrable to other spheres of pathology.

The authors earlier developed and recently reported a method to create 3D MRI-CT fusion images of the OCD lesions. It his article authors describe a computer-aided technique that combines the advantages of CT and MRI and provides a minimally invasive, accurate operative evaluation of OCD lesions. In addition, a detailed surgical simulation is possible, which could aid surgeons in intraoperative decision-making.

Section of Materials and Methods is very easy to understand and to follow all the steps of the presented research (1.Obtaining the MR and CT images, 2. Creation of 3D models, 3. Fusion of created 3D models, 4. Lesion evaluation using MCFI, 5. Surgical simulation).

The Results of the simulation are corresponding to ICRS Cartilage Injury Evaluation Package.

A detailed surgical simulation based on the newly proposed model is potentially useful for minimizing intraoperative decisionmaking and thus minimizing the risk to the patient.

Author Response

Authors’ Reply to the Review Report (Reviewer 2)

Reviewer 2

We would like to thank the reviewer for his/her thoughtful comments. Based on the revision comments, we believe that no revision is required.

Osteochondritis dissecans (OCD) is a specific kind of pathology and no single imaging modality can adequately predict lesion severity thereby presenting a challenge to elbow surgeons. Though the authors address a rare pathology, the overall results could be transferrable to other spheres of pathology.

The authors earlier developed and recently reported a method to create 3D MRI-CT fusion images of the OCD lesions. It his article authors describe a computer-aided technique that combines the advantages of CT and MRI and provides a minimally invasive, accurate operative evaluation of OCD lesions. In addition, a detailed surgical simulation is possible, which could aid surgeons in intraoperative decision-making.

Section of Materials and Methods is very easy to understand and to follow all the steps of the presented research (1.Obtaining the MR and CT images, 2. Creation of 3D models, 3. Fusion of created 3D models, 4. Lesion evaluation using MCFI, 5. Surgical simulation).

The Results of the simulation are corresponding to ICRS Cartilage Injury Evaluation Package.

A detailed surgical simulation based on the newly proposed model is potentially useful for minimizing intraoperative decisionmaking and thus minimizing the risk to the patient.

⇨ Thank you for your thoughtful comments. We are glad to see these positive comments. We hope that our manuscript meets the journal’s aims and scope.